# Depressed mood as a transdiagnostic target relevant to anxiety and/or psychosis: a scoping review protocol

Jermaine Dambi ,[1,2] Edwin Mavindidze ,[1] Primrose Nyamayaro,[3] Rhulani Beji-Chauke,[2] Tariro Dee Tunduwani,[1] Beatrice K Shava,[1,2] Webster Mavhu ,[4,5] Melanie Abas,[6] Dixon Chibanda,[2,3,7] Clement Nhunzvi [1,8]

JD and EM are joint first authors.

For numbered affiliations see end of article.

**Correspondence to**
Dr Jermaine Dambi;
jermainedambi@gmail.com

## ABSTRACT

**Introduction** Depressed mood is a psychological state characterised by sadness or loss of interest in activities. Depressed mood is a highly prevalent symptom across major mental disorders. However, there is limited understanding of the burden and management of comorbid depressed mood across major mental disorders. Therefore, this scoping review aims to summarise knowledge on depressed mood among persons with anxiety and/or psychosis. The specific aims are to describe the epidemiology and risk factors of depressed mood as a transdiagnostic target among persons with anxiety and/or psychosis, to identify commonly used outcome measures for depressed mood and to outline initial evidence of psychometric robustness and to identify and summarise the effectiveness of commonly applied depressed mood modification interventions. Our hope is that the proposed review will provide insights into the burden of depressed mood in persons with anxiety and psychosis and help to identify evidence gaps and recommendations for future research.

**Methods and analysis** This scoping review will be conducted per Arksey and O'Malley's framework. We will first search for peer-reviewed articles and grey literature published from 2004 to 2023 in PubMed, Scopus, Web of Science, Africa-Wide Information, CINAHL, PsycINFO, Academic Search Premier, Humanities International Complete, Sabinet, SocINDEX, Open Grey and Google Scholar. We will include articles reporting depressed mood (subthreshold depression) among persons with anxiety and/or psychosis. Studies recruiting participants meeting depression diagnostic criteria and those published in non-English languages will be excluded. Two independent researchers will extract the data. We will analyse and chart data collaboratively with researchers with lived experiences of depressed mood.

**Ethics and dissemination** This study does not require ethical approval as it is a literature review. The results will be submitted for publication in a peer-reviewed journal.

## STRENGTHS AND LIMITATIONS OF THIS STUDY

⇒ Use of robust scoping review framework.
⇒ Duplicate article screening and data collection.
⇒ Involvement of persons with lived experiences of depressed mood.
⇒ Possibility of language bias.

## INTRODUCTION

A depressed mood is a temporary psychological state characterised by sadness, irritability, emptiness, unhappiness or loss of pleasure/interest in activities.[1 2] Although depressed mood can sometimes lead to depression or other mental disorders, it differs from clinical depression as it is a fleeting psychological state that can resolve naturally without treatment.[3] Depressed mood is a highly prevalent early sign and symptom across major mental disorders, including mood disorders (eg, depression and bipolar affective disorders), psychosis, anxiety disorders, substance use disorders and personality disorders.[1 2 4] A recent meta-analysis showed the odds of depressed mood in persons with generalised anxiety disorders to be up to 12 times (95% CI 5.2 to 26.3) as compared with persons without generalised anxiety disorders.[2] Also, up to 23% of persons experiencing first-episode psychosis suffer from depression.[1]

Diagnosing mental disorders is based on observations of latent constructs (signs and symptoms). The exact aetiology of depressed mood in patients with comorbid mental disorders such as anxiety and psychosis is unknown; several theories have been postulated.[4 5] For example, the network theory of mental disorders posits that comorbidity is due to concuring symptoms.[5] Coexisting symptoms have a compound effect, that is, the more significant the co-occurrence, the greater the morbidity.[1 5] For instance, the association between depression and anxiety is bidirectional, that is, they are mutual bidirectional risk factors.[2 4] Patients with depression are at increased risk of developing anxiety, with the reverse also true. Depressed mood and psychosis are also mutual bidirectional

risk factors.[1] In persons with psychosis, comorbid anxiety/depression is often associated with severe symptoms, poor prognosis and increased psychotic experiences.[1] The high prevalence of coexisting symptoms across mental disorders has inherently led to an inclination towards the utilisation of transdiagnostic interventions.

The transdiagnostic treatment approach is premised on interventions designed to treat co-occurring symptoms simultaneously.[6] The transdiagnostic treatment approach can mitigate the enormous mental healthcare gap by addressing depressed mood and several mental disorders. Globally, there is a substantial mental healthcare gap, that is, there is a huge mismatch in the high number of people in need of mental healthcare against the few available treatment/care resources.[7] The care gap spans all mental disorders, including depressed mood, which is highly prevalent.[8 9] Consequently, transdiagnostic interventions can potentially mitigate the care gap as multimodal interventions, such as the WHO Mental Health Gap Action Programme, have been successfully implemented to address a wide array of mental disorders.[7] Also, Wellcome Trust has coined the term transdiagnostic target to imply any co-occurring symptoms and experiences considered critical by people with lived experiences as 'best bets' for intervention with a high potential return-on-investment and downstream improved health-related quality of life (HRQoL) and functioning in people with mental health disorders.[10] Of the multiple mental disorder symptoms, depressed mood is a significant transdiagnostic target with multilevel impacts. Depressed mood, a precursor to depression and other mental conditions, often leads to cognitive difficulties (eg, poor attention and concentration), emotional exhaustion, feelings of excessive guilt or low self-esteem, sleep challenges, and suicidal thoughts; classical depression signs and symptoms.[4 5] The depressed mood also accounts for physical challenges accompanying mental illnesses, such as changes in appetite or weight, tiredness, low energy and disrupted sleeping patterns.[2 11] Depressed mood, therefore, impairs the achievement of functional and personal goals of those affected, negatively affecting their HRQoL.[4] Lastly, comorbid depressed mood predicts severe morbidity, substance misuse and treatment resistance.[2 4 5]

Given the multilevel effects of depressed mood, especially if allowed to progress, early detection, referral and management are critical. Early screening is contingent on the availability of validated screening tools. Depressed mood is measurable, and several screening tools are available, but their evidence of diagnostic accuracy is fragmented. For example, the Patient Health Questionnaire (PHQ-9) is one of the most commonly used tools, with evidence of diagnostic accuracy and psychometric performance across systematic reviews.[12 13] The PHQ-9, like other screening tools, has been used globally across different contexts (clinical or research) with comparable transcultural validity and diagnostic performance.[12 13] Screening tools are often used conjunctively with clinical diagnostic interviews to ascertain the clinical diagnosis

and guide treatment. However, due to human resources shortages, screening results can inform treatment without a confirmatory clinical diagnostic interview.

Fortunately, there are scalable, effective, low-cost interventions for managing depressed mood across socioeconomic contexts. For example, Friendship Bench is an evidence-based, task-shifting intervention effective in managing depressed mood.[14] The Friendship Bench has been successfully implemented in a low-income country (Zimbabwe) and is being scaled up globally.

Furthermore, physical modalities such as exercise have been proven to be effective in mitigating the effects of depressed mood across all age groups.[15] For instance, a recent meta-analysis showed that exercise had moderate effects on depressed mood.[16] Therefore, interventions that include exercise can be a low-hanging fruit that can potentially reduce the progression of depressed mood. Early interventions targeting depressed mood can have far-reaching and sustainable outcomes. As with other mental conditions/psychological states, early treatment should presumably increase treatment engagement, resulting in improved HRQoL, increased functioning and community participation.[1 2 4] Managing depressed mood should give the affected person a fair chance to engage and benefit from other treatments requiring active participation and increased pleasure and interest in activities. However, there is limited understanding of the burden and management of comorbid depressed mood across major mental disorders. Therefore, this review aims to summarise knowledge on depressed mood as a transdiagnostic target among persons with anxiety and/or psychosis. The specific aims are to:

1. Describe the epidemiology and risk factors of depressed mood as a transdiagnostic target among persons with anxiety and/or psychosis.
2. Identify commonly used outcome measures for depressed mood and outline initial evidence of psychometric robustness.
3. Identify and summarise the effectiveness of commonly applied depressed mood modification interventions.

Our hope is that the proposed review will provide insights into the burden of depressed mood in persons with anxiety and psychosis and help to identify evidence gaps and recommendations for future research.

## METHODS AND ANALYSIS
### Overview
We will conduct a scoping review (SR) to review and summarise the current knowledge base on depressed mood among those with anxiety and/or psychosis. The SR will be conducted per Arksey and O'Malley's framework.[17] An SR aims to identify the available evidence on a particular concept or subject.[18] The review is reported Preferred Reporting Items for Systematic Reviews and Meta-Analyses for Scoping Reviews[19] (see online supplemental file 1). Below, we describe the SR methodology

guided by the six stages of Arksey and O'Malley's framework.

## Stage 1: identifying the research question(s)

In the context of depressed mood, this scoping aims to seek to answer these questions:

1. What are the epidemiology and risk factors of depressed mood in persons with anxiety and psychosis?
2. What are commonly used outcome measures used for depressed mood in persons with anxiety and psychosis?
3. What is the collective evidence of psychometric robustness of commonly used depressed mood outcome measures in persons with anxiety and psychosis?
4. What approaches and interventions are used to modify depressed mood in persons with anxiety and psychosis?
5. In persons with anxiety and psychosis, how effective are commonly applied depressed mood modification interventions?

The research team engaged in an extensive and iterative process of consultation and interactions with various stakeholders, including persons with lived experience of anxiety and/or psychosis, to refine the review questions further. Once the research questions are finalised, the team will refine keywords that will guide the final search strategy.

## Stage 2: identifying relevant studies

The goal is to search broadly enough to capture all the available evidence that answers the study question by producing meaningful results.[20] Through iterative cycles of preliminary searches and refinement, the team developed the inclusion criteria outlined below.

### Study inclusion and exclusion criteria

For this review, we will include:

► Peer-reviewed articles, grey literature, original research and conceptual papers will be included. There will be no restrictions on the type of study designs and settings included to attain literature saturation.
► We will include literature published in English. We do not have the financial resources to translate and analyse studies published in other languages.
► Literature published from 2004 to 2023. The year 2004 will be selected as the start date as quick pilot searches show it to be the year the first article on transdiagnostic targets was published.

Literature that reports on the identification and/or management of comorbid depressed mood in patients with anxiety and or psychosis. We will include anxiety disorders per The Diagnostic and Statistical Manual of Mental Disorders, Fifth Edition (DSM 5) criteria, including generalised anxiety, panic disorder, social anxiety disorder and specific phobias. We excluded anxiety disorders due to medications or other medical conditions and obsessive–compulsive disorders. For psychosis, we will include those meeting the diagnostic criteria of psychotic disorder as specified in DSM-5, characterised by delusions,

**Table 1** CINHAL and PubMed search strategy

| Key search terms | Alternative terms | PubMed |
|---|---|---|
| Depressed mood | Depressive symptoms OR Depressive affect OR Depress* | Depression [MeSH] |
| Transdiagnostic | | Transdiagnostic |
| Psychosis | Psychotic disorder OR Psychoses OR Psycho* OR Schizophreni* (For schizophreniform, Schizophrenia) Schizoaffective OR Brief reactive psychosis OR Substance-induced psychoses | Psychotic Disorders [MeSH] OR Psychoses OR Substance-induced [MeSH] |
| Anxiety | Social anxiety OR Nervousness OR Anxiousness OR Generalised anxiety disorder OR Panic attack OR phobia Anxiety Disorder OR Anxi* OR Neurotic OR Neuroses | Anxiety [MeSH] OR Anxiety Disorders [MeSH] |

hallucinations, disorganised thinking and negative symptoms. Psychosis diagnoses include schizophrenia, delusional disorders, reactive psychosis, substance-induced psychosis, schizoaffective and schizophreniform. We will similarly exclude psychotic conditions due to neurological or medical conditions. Also, literature focusing on subthreshold anxiety and or psychosis will be excluded. We also aim to exclude those studies focusing on clinical depression.

### Search strategy

Key search terms included "depressed mood and its variants", "transdiagnostic", and "psychosis and its variants", and "anxiety and its alternative terms". We applied a combination of Boolean logic operators to glean the literature. Table 1 outlines an example search strategy for the CINHAL and PubMed databases. Online supplemental file 2 outlines the search strategy for the rest of the databases.

### Databases

Utilising keywords, two researchers (JD and EM) will conduct broad searches in these electronic databases: PubMed, Academic Search Premier, Africa-Wide Information, the Cumulative Index to Nursing and Allied Health Literature (CINAHL), Humanities International Complete, PsycINFO, Sabinet, SocINDEX, Scopus, Web of Science, Open Grey and Google Scholar. These databases were chosen based on expert advice and their relevance

to the research questions. Guided by an expert librarian, we will conduct preliminary searches in PubMed, refine key search terms and identify MeSH terms. We will then further customise the search strategy for other databases. Online supplemental file 2 is an outline of the search strategy for the rest of the databases.

### Stage 3: study selection
#### Data management
Once the literature search is complete, the citations will be uploaded to Rayyan[21] to prepare for the study selection process.

#### Screening
Two review teams (team 1: JD and PM; team 2: RB-C and EM) will screen the retrieved articles in three steps. First, both teams independently review citations to identify articles for inclusion. The articles will also be assessed to determine whether they meet the selection criteria at both the title and abstract levels. At this preliminary stage, the goal is to identify articles focusing on depressed mood as a transdiagnostic target. If there is uncertainty, the article was further reviewed in the full article screening step. The included articles will be captured on an Excel spreadsheet, and the lists from the two teams compared. In cases of reviewers' disagreement(s), a consensus meeting will be convened, where the article will be further reviewed to make the final decision. Full texts of included studies will be retrieved to prepare for data extraction. Last, we will not perform any quality appraisal of the identified studies. This review aims to landscape and summarise the available evidence on depressed mood, as previous reviews have exclusively focused on depression.

### Stage 4: charting the data
Two researchers not involved in the screening process (CN and WM) will independently review and read the full articles and rate them against the inclusion criteria. Where reviewers do not agree, a consensus meeting will be held to agree on a decision. Throughout all three steps, Cohen's kappa will be calculated to assess the level of agreement between reviewers. If the level of agreement is weak ($\kappa < 0.60$), the process is repeated until there is a strong agreement. This stage aims to contextualise the study findings by briefly describing each article, including the methodology used to reach conclusions.[17] Bibliometric information, including the authors, year of publication, country of study, the study design, study population and essential results, will be captured in Microsoft Excel.

### Stage 5: collating, summarising and reporting the results
Consistent with an SR approach, the extracted data will be collated and presented in a narrative synthesis applying descriptive statistical (tabular supplements) and qualitative methods.[22] The results will be presented thematically in line with the study objectives. We will qualitatively summarise the epidemiology of transdiagnostic depressed mood, its outcome measures, interventions and natural progression in anxiety and psychosis. For instance, through numerical analysis, we will produce charts and tables highlighting the distribution of the findings concerning geographical distribution, distribution across diagnoses, distribution across the years and the nature of the reported studies. The scoping nature of the study requires a qualitative approach. To further illustrate this, we will not perform a meta-analysis to synthesise risk factors associated with depressed mood by calculating a summative effect size; instead, we will qualitatively describe the most reported risk factors.

### Stage 6: consultation exercise
Before compiling a final report, we will present our preliminary findings to various stakeholders, including individuals with lived experience, clinicians and researchers. The research team will also orient the stakeholders on the SR objectives before presenting the preliminary findings. This will help gain more insight into the meaning of the results and guide the research team towards resources and evidence that might have been missed through the search strategy in preparation for the final report. Additionally, this stage will serve as a precursor to disseminating study findings.

### Patient and public involvement
The experiential meaning and impact of depressed mood as a transdiagnostic target can be further explored through the narratives of those with lived experience. We will conduct extensive and iterative processes of consultation and interaction with various stakeholders, including persons with lived experience of anxiety and/or psychosis, to refine the review questions. We will also triangulate insights from our SR with contextual data from key informants, including persons with lived experience, clinicians and researchers. We will apply purposive sampling to reach information-rich informers and engage in two rounds of focus group discussions and in-depth interviews.

### ETHICS AND DISSEMINATION
This study does not require ethical approval as it is a literature review. The results will be submitted for publication in a peer-reviewed journal. All relevant data/materials, including data collection tools, will be submitted as online supplemental files when the results are submitted for publication.

**Author affiliations**
[1]Department of Rehabilitation Sciences, University of Zimbabwe Faculty of Medicine and Health Sciences, Harare, Zimbabwe
[2]Friendship Bench, Harare, Zimbabwe
[3]Department of Primary Health Care Sciences, Unit of Mental Health, University of Zimbabwe, Harare, Zimbabwe
[4]Centre for Sexual Health and HIV/AIDS Research Zimbabwe, Harare, Zimbabwe
[5]Liverpool School of Tropical Medicine, Liverpool, UK
[6]Department of Health Service and Population Research, King's College London, London, UK
[7]London School of Hygiene & Tropical Medicine, London, UK

[8]Bond University Faculty of Health Sciences and Medicine, Gold Coast, Queensland, Australia

**Contributors** JD and EM were primarily responsible for writing the first draft of the protocol. JD, EM, PN, RB-C, TDT, BKS, MA, DC, WM and CN were involved in conceptualising the study and editing all protocol manuscript versions. JD will search the literature and data management. EM, PN, RB-C, TDT, BKS, MA, DC, WM and CN will be responsible for article screening, quality assurance, data extraction and qualitative synthesis. MA, WM, CN and DC will provide overall supervision, mentoring and guidance.

**Funding** Wellcome Trust (grant number NA).

**Competing interests** None declared.

**Patient and public involvement** Patients and/or the public were involved in the design, or conduct, or reporting, or dissemination plans of this research. Refer to the Methods section for further details.

**Patient consent for publication** Not applicable.

**Provenance and peer review** Not commissioned; externally peer reviewed.

**ORCID iDs**
Jermaine Dambi http://orcid.org/0000-0002-2446-7903
Edwin Mavindidze http://orcid.org/0000-0001-9849-8932
Webster Mavhu http://orcid.org/0000-0003-1881-4398
Clement Nhunzvi http://orcid.org/0000-0001-5804-9817

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
