## [Reviewer comments · BMJ Open]

ARTICLE DETAILS

TITLE (PROVISIONAL)	Depressed mood as a transdiagnostic target relevant to anxiety and/or psychosis: a scoping review protocol
AUTHORS	Dambi, Jermaine; Mavindidze, Edwin; Nyamayaro, Primrose; Beji-Chauke, Rhulani; Tunduwani, Tariro; Shava, Beatrice K.; Mavhu, Webster; Abas, Melanie; Chibanda, Dixon; NHUNZVI, CLEMENT

VERSION 1 – REVIEW

REVIEWER	Martin-Villalba, Ines University of Barcelona, Department of Clinical Psychology and Psychobiology
REVIEW RETURNED	31-Oct-2023

GENERAL COMMENTS	In reference to the objective "Describe the epidemiology and risk factors of depressed mood as a transdiagnostic target among persons with anxiety and/or psychosis", please specify which diagnoses are included within the diagnostic categories anxiety and psychosis.
---

REVIEWER	Stafford, Jean University College London, Lifelong Health and Ageing
REVIEW RETURNED	19-Jan-2024

GENERAL COMMENTS	This protocol for a scoping review is clearly described and addresses an important and under-researched topic, following an appropriate framework. My comments are as follows: • The concept of transdiagnostic targets is mentioned in paragraph three of the introduction, but both transdiagnostic symptoms and transdiagnostic interventions require further explanation and elaboration.• In the introduction, only one low-cost intervention is described for depressed mood. This section should be expanded to briefly summarise other forms of intervention, with further examples.• More clarity is needed in describing whether research questions 2-5 refer to outcome measures and interventions for depression in the context of anxiety and psychosis only, or to depression in general.• The mental health care gap is briefly mentioned in the introduction. This concept requires a brief explanation, with further information about how transdiagnostic interventions could help to address this.• In the introduction, depressed mood is described as leading to cognitive difficulties and accounting for physical challenges. It may be more appropriate to refer to depression being associated with these factors.
--

	 • The inclusion criteria should clarify whether studies focused on subthreshold anxiety and psychosis symptoms be included, as well as diagnosed anxiety and psychotic disorders. Similarly, although the focus is on depressed mood, inclusion criteria should state whether diagnosed depression will also be considered. Search terms should also reflect this. I also wondered whether obsessive compulsive disorder and post-traumatic stress disorder will be considered within the scope of the review. • Will there be any restrictions on the type of study designs included (e.g. cross-sectional and longitudinal) and settings (for instance will you include studies in clinical settings) to characterise the epidemiology and risk factors? • A narrative synthesis is mentioned, but more detail is needed about the approach that will be used. • The information listed in the ethics and dissemination section is not directly relevant to the subheading, as it describes possible implications of the findings. The subheading should be changed, and/or this section should provide additional details on how findings will be disseminated. • Will the quality of individual studies be critically appraised? If so, further details about the approach to appraisal should be provided. • Further information about the composition of the patient and public involvement group would be beneficial, and the authors should clarify whether the group will be involved in other aspects of the study, such as commenting on the findings and their implications, as well as guiding dissemination plans. Minor points  • There is an error in the first bullet point under 'Strengths and Limitations' within the abstract, where 'sscoping review' should be corrected to 'scoping review.' • In line 8, the term 'mental conditions' could be reworded, perhaps to 'mental disorders' for consistency. • In lines 12-13, a reference group should be mentioned when describing the odds of depressed mood being higher in those with generalised anxiety disorders (e.g. relative to those without anxiety). • In line 26, the phrase 'great propensity towards' needs rephrasing for better clarity. • In line 40, the term 'only' in 'only contingent upon' can be omitted. • In the methods section, Stage 1, point 2, the word 'used' is repeated twice.
--	--

VERSION 1 – AUTHOR RESPONSE

Q u e r y #	Section	Reviewer #	Comment	Response	Reference
1.		1	In reference to the objective "Describe the epidemiology and risk factors of depressed mood as a transdiagnostic target among persons with anxiety and/or psychosis", please specify which diagnoses are included within the diagnostic categories anxiety and psychosis.	Thank you for your comment. We will include anxiety disorders per DSM 5 criteria, including, generalised anxiety, panic disorder, social anxiety disorder, and specific phobias. We excluded, anxiety disorders due to medications or other medical conditions and obsessive-compulsive disorders. For psychosis, we will include those meeting the diagnostic criteria of psychotic disorder as specified in DSM 5 characterised by delusions, hallucinations, disorganised thinking, and negative symptoms. Psychosis diagnoses include schizophrenia, delusional disorders, reactive psychosis, substance-induced psychosis, schizoaffective, and schizophreniform. We will similarly exclude psychotic conditions due to neurologic or medical conditions.	Line 138-144
		2	In line 8, the term 'mental conditions' could be reworded, perhaps to 'mental disorders' for consistency.	Thank you for your comment, we have made the necessary changes. The line now reads " mental disorders".	Line 78
		2	In lines 12-13, a reference group should be mentioned when describing the odds of depressed mood being higher in those with generalized anxiety disorders (e.g. relative to those without anxiety).	Thank you for the comment, we have added the phrase "as compared to persons without generalized anxiety disorders"	Line 83-84
		2	In line 26, the phrase 'great propensity towards' needs rephrasing for better clarity.	Thank you for pointing out this ambiguity, we have amended the phrase which now reads " an inclination towards utilising..."	Line 98

		2	In line 40, the term 'only' in 'only contingent upon' can be omitted.	Thank you for your comment, we have made the necessary changes.	Line 113
5		2	The concept of transdiagnostic targets is mentioned in paragraph three of the introduction, but both transdiagnostic symptoms and transdiagnostic interventions require further explanation and elaboration	Thank you for the comments. We have defined the concepts as suggested. The amended statements now read: Transdiagnostic interventions focuses on identifying and targeting multiple maladaptive broad arrays of diagnostic presentations of common mental disorders in treatment [9]. Recent meta-analysis which implemented transdiagnostic interventions i.e. CBT have demonstrated high efficacy levels in significantly reducing levels of both anxiety and depressed mood[10]	
6		2	In the introduction, only one low-cost intervention is described for depressed mood. This section should be expanded to briefly summarise other forms of intervention, with further examples.	Thank you for your comment. We have added statements that read "Furthermore, physical modalities such as exercise have been proven to be effective in mitigating the effects of depressed mood across all age groups [12]. For instance, a recent meta-analysis showed that exercise showed moderate effects on depressed mood".	Lines 126-128.
7		2	More clarity is needed in describing whether research questions 2-5 refer to outcome measures and interventions for depression in the context of anxiety and psychosis only, or to depression in general	Thank you, for the suggestions. To improve clarity, we have clearly articulated the PICO elements by defining the population, interventions, and outcomes as appropriate. Kindly refer to the amended research questions. we have added statements to further clarify	Lines 155-162
8		2	The mental health care gap is briefly mentioned in the introduction. This concept requires a brief explanation, with further information about how transdiagnostic interventions could help to address this	Thank you for the important suggestion. We have briefly described the concept of mental care gap and the utility of transdiagnostic interventions. The statement now reads, "several mental disorders by specifically targeting co-occurring symptoms [10]. Globally, there is a huge mental health care gap i.e. there is a huge mismatch in the high number of people in need of mental healthcare against the few available treatment/care resources [8]. The care gap spans across all mental disorders, including depressed mood, which is highly prevalent [6,7]. Consequently, transdiagnostic interventions can potentially mitigate the care gap as multimodal interventions such as the WHO Mental	

				Health Gap Action Programme, have been successfully implemented to address a wide array of mental disorders [8].”	
9		2	In the introduction, depressed mood is described as leading to cognitive difficulties and accounting for physical challenges. It may be more appropriate to refer to depression being associated with these factors.	Thank you for pointing out this ambiguity. to improve clarity, we have edited the statement to, “Depressed mood, a precursor to depression and other mental disorders, leads to cognitive difficulties (e.g. poor attention and concentration), emotional exhaustion, feelings of excessive guilt or low self-esteem, sleep challenges, and suicidal thoughts; classical depression signs and symptoms [4,5].”	Lines
10	Methods and Materials	Editor in Chief	Thank you for including the CINHAL and PubMed search strategy in table 1. In addition, please include, as a supplementary file, the precise, full search strategies for all databases, registers and websites, including any filters and limits to be used. This supplementary file should be cited in the main text (eg, where you indicate “We further customised the search strategy for the other databases”).	We have added the supplementary file as a separate document. Also, we have added this text with the main text document, “Additional File 1 is an outline of the search strategy for the rest of the databases.”	
11		2	In the methods section, Stage 1, point 2, the word 'used' is repeated twice.	Thank you for highlighting that, we have omitted the repeated word.	Lines 122 – 125.
12		2	The inclusion criteria should clarify whether studies focused on subthreshold anxiety and psychosis symptoms be included, as well as diagnosed anxiety and psychotic disorders. Similarly, although the focus is on depressed mood, inclusion criteria should state whether diagnosed depression will also be considered. Search terms should also reflect this. I also wondered whether obsessive compulsive disorder and post-traumatic stress disorder will be considered within the scope of the review.	Thank you for highlighting that, we have clarified that in the subsequent section. Essentially, we will exclude studies focusing on sub-threshold anxiety and or psychosis will be excluded. We also aim to exclude those studies focusing on clinical depression as this is the gap the review aims to fill.	Lines 173-184
13		2	Will there be any restrictions on the type of study designs included (e.g. cross-sectional and longitudinal) and settings	Thank you for highlighting this ambiguity, however, there will be no restrictions on the study designs included. This will enable us to thoroughly scope for available literature that can add to our work’s body of knowledge. We have	Lines 173.

			(for instance will you include studies in clinical settings) to characterise the epidemiology and risk factors?	added the following statement, "There will be no restrictions on the type of study designs and settings included to attain literature saturation".	
1 4		2	A narrative synthesis is mentioned, but more detail is needed about the approach that will be used.	Thank you for your comment, we have re-written this section to improve clarity by describing that the narrative synthesis will be in form of descriptive statistical summative methods and qualitative approaches to evidence synthesis.	
1 5		2	The information listed in the ethics and dissemination section is not directly relevant to the subheading, as it describes possible implications of the findings. The subheading should be changed, and/or this section should provide additional details on how findings will be disseminated.	Thank you for your comment, we have moved the ethics and dissemination section to the abstract and relabelled this section as implications and dissemination".	Lines 61-65
1 6		2	Will the quality of individual studies be critically appraised? If so, further details about the approach to appraisal should be provided.	Thank you for pointing this out. We have added the following statement to improve clarity, "Last, we will not perform any quality appraisal of the identified studies as this review aims to landscape and summarise the available evidence on depressed mood as previous reviews have exclusively focused on depression."	

VERSION 2 – REVIEW

REVIEWER	Stafford, Jean University College London, Lifelong Health and Ageing
REVIEW RETURNED	10-May-2024
GENERAL COMMENTS	I am satisfied that the authors have addressed my comments.